# Effects of Biological/Targeted Therapies on Bone Mineral Density in Inflammatory Arthritis

**DOI:** 10.3390/ijms23084111

**Published:** 2022-04-08

**Authors:** Tai-Li Chen, Kai-Hung Chang, Kuei-Ying Su

**Affiliations:** 1Department of Medical Education, Hualien Tzu Chi Hospital, Buddhist Tzu Chi Medical Foundation, Hualien 970, Taiwan; terrychen.a@gmail.com; 2Center for Aging and Health, Hualien Tzu Chi Hospital, Buddhist Tzu Chi Medical Foundation, Hualien 970, Taiwan; 3Division of Plastic Surgery, Department of Surgery, Hualien Tzu Chi Hospital, Buddhist Tzu Chi Medical Foundation, Hualien 970, Taiwan; 102311131@gms.tcu.edu.tw; 4Division of Allergy, Immunology and Rheumatology, Hualien Tzu Chi Hospital, Buddhist Tzu Chi Medical Foundation, Hualien 970, Taiwan; 5School of Medicine, Tzu Chi University, Hualien 970, Taiwan

**Keywords:** inflammatory arthritis, osteoporosis, bone-impacting drugs

## Abstract

Inflammatory arthritis has been reported to be associated with the development of osteoporosis. Recent research has investigated the mechanisms of bone metabolism in chronic inflammatory arthritis such as rheumatoid arthritis (RA) and spondyloarthritis (SpA). Progress in both animal and clinical studies has provided a better understanding of the osteoclastogenesis-related pathways regarding the receptor activator of nuclear factor-κB ligand (RANKL), anti-citrullinated protein antibodies (ACPAs), and Wnt signaling and Dickkopf-related protein 1 (Dkk-1). The complex interplay between inflammatory cytokines and bone destruction has been elucidated, especially that in the interleukin-17/23 (IL-17/23) axis and Janus kinase and signal transducer and activator of transcription (JAK-STAT) signaling. Moreover, advances in biological and targeted therapies have achieved essential modifications to the bone metabolism of these inflammatory arthritis types. In this narrative review, we discuss recent findings on the pathogenic effects on bone in RA and SpA. Proinflammatory cytokines, autoantibodies, and multiple signaling pathways play an essential role in bone destruction in RA and SpA patients. We also reviewed the underlying pathomechanisms of bone structure in biological and targeted therapies of RA and SpA. The clinical implications of tumor necrosis factor inhibitors, abatacept, rituximab, tocilizumab, Janus kinase inhibitors, and inhibitors of the IL-17/23 axis are discussed. Since these novel therapeutics provide new options for disease improvement and symptom control in patients with RA and SpA, further rigorous evidence is warranted to provide a clinical reference for physicians and patients.

## 1. Introduction

Osteoporosis is associated with inflammatory arthritis, which has had an increasing global prevalence in recent decades [1]. Patient care regarding osteoporosis requires integrated interdisciplinary care [1]. Bone involvement is the hallmark of several inflammatory arthritis types, such as rheumatoid arthritis (RA) and spondyloarthritis (SpA) [2,3]. There are three major forms of bone loss: (a) focal bone erosions at the joint margins, (b) periarticular osteolysis and new bone formation adjacent to the inflamed joints, and (c) systemic bone loss (osteoporosis). The influence of RA and SpA on bone metabolism and arthritic pathology can lead to arthralgia, joint swelling, and limited physical activity. As these bone involvements significantly affect patients’ quality of life, the underlying mechanisms of bone loss in chronic inflammatory arthritis have driven both laboratory and clinical discussions. Previous studies have investigated bone remodeling in patients with inflammatory arthritis [4,5]. The pathophysiology of bone remodeling is complex, including interactions between the inflammatory and immune systems. The associations of inflammatory cytokines with osteocytes and bone resorption are also complicated.

As the understanding of the pathophysiology of RA and SpA has increased, numerous biological/targeted therapies have been studied and approved for clinical applications. Recent studies have investigated their effects on bone mineral density and bone metabolism, which may also impact clinical symptoms and patients’ quality of life. In this narrative review, we provide an overview of the pathophysiology of RA and SpA. We primarily focused on the effect of targeted therapies, including biologics and small molecule inhibitors, on bone mineral density and osteoporosis in inflammatory arthritis. In Figure 1, we summarize the pathophysiology discussed in this article.

## 2. Rheumatoid Arthritis

### 2.1. Bone Resorption and Osteoporosis in Rheumatoid Arthritis

RA is a chronic immune-mediated disease affecting about 5 in 1000 people worldwide [6]. The development of RA involves a complex interplay among the immune system, gene predisposition, and environmental triggers [7,8]. Clinically, RA is characterized by articular bone erosion, progressive joint destruction, and deformities, resulting in disability [2,9]. Of note, RA increases the risk of osteoporosis and joint cartilage destruction [9]. Clinical evidence has also supported an increased fracture risk in patients suffering from RA [9]. Indeed, examinations using high-resolution peripheral quantitative computed tomography (HRpQCT) have found that RA patients are associated with increased cortical porosity and reduced bone strength, representing higher risks of fractures than healthy controls [10]. Recent studies have investigated the underlying pathogenesis of bone resorption in RA, which is discussed below.

#### 2.1.1. Proinflammatory Cytokines and Synovitis

Inflammation in RA is mainly driven by cytokines, including tumor necrosis factor-α (TNF-α), interleukin-6 (IL-6), and interleukin-1 (IL-1) [11]. The involved proinflammatory cytokines can directly or indirectly provoke osteoclast activation and halt osteoblast differentiation, leading to bone loss and subsequent osteoporosis [12,13]. Cytokines such as interleukin (IL)-1β, IL-6, IL-8, and IL-11 have been reported to be directly associated with osteoclastogenesis. Other cytokines such as TNF-α, IL-7, and IL-15 may indirectly affect the formation of osteoclasts or the inhibition of osteoblasts. In addition, chronic inflammation can lead to osteoporosis through the release of matrix metalloproteinases (MMPs) [14]. The receptor activator of nuclear factor-κB ligand (RANKL) is one of the major cytokines that could be modulated in the pathogenesis of RA bone loss and cartilage damage [7,15]. In patients with RA, the primary source of RANKL is synovial fibroblasts and CD4 + CD28- T -cells. Moreover, RANKL was found to produce both a positive effect on osteoclastogenesis and a detrimental effect on osteoblastic development [16]. A complicated interplay between multiple proinflammatory cytokines and RANKL-associated osteoclastogenesis is considered to have clinical effects on osteolysis. 

#### 2.1.2. Autoantibodies

Anti-citrullinated protein antibodies (ACPAs) are the most specific serological biomarkers not only for predicting the development of RA but also for disease prognosis [17,18]. ACPAs are mainly produced by plasmablasts and plasma cells [17,18]. Recent animal studies have suggested that ACPAs may stimulate osteoclast differentiation and initiate bone change [19]. ACPAs enhance osteoclast differentiation from monocyte-derived or circulating CD1c+ dendritic cells (DCs) by increasing the release of IL-8 [20]. ACPAs binding to immature DCs might be associated with the activation and differentiation toward the osteoclast lineage, facilitating bone erosion in ACPA-positive RA [21,22]. Interestingly, immunization with citrullinated Type II mouse collagen resulted in increased ACPA levels and lowered bone quality, but these were uncoupled from the degree of inflammation [23]. Recent advances in mass spectrometry have even proposed insights into a term called “citrullinome” in RA [24]. In in vitro models, ACPAs isolated from RA patients had the potential to stimulate both murine and human osteoclast precursor cells [22,25,26]. ACPAs transferred to mice had the ability to bond to bone marrow-resident osteoclasts and osteoclast precursor cells, leading to joint pain and bone erosion [22,23]. This mechanism of bone resorption could explain why bone destruction in RA progresses with the existence of inflammation (in healthy ACPA-positive individuals or patients achieving sustained clinical remission) [22,25]. Further analyses have identified extracellular IL-8 as the key mediator of ACPA-triggered osteoclastogenesis and bone remodeling in mouse models [23,26]. Most importantly, blocking IL-8 activity could reverse the pathogenic effects of ACPAs both in vitro and in vivo [22,25,26]. Future translational research is needed to clarify its clinical importance.

#### 2.1.3. The Wnt Signaling Pathway and Dickkopf-Related Protein 1 (Dkk-1)

Wnt proteins are palmitoylated and glycosylated ligands that have a pivotal role in the regulation of bone remodeling [27]. Wnt signaling in osteoblasts regulates the expression of RANKL [28]. Furthermore, Wnt/β-catenin signaling directly affects the regulation of osteoclastogenesis [29,30,31]. The Wnt and the osteoprotegerin (OPG)-RANKL-RANK signaling systems, as critical mediators, interact in subchondral bone remodeling [32,33]. Dkk-1, a Wnt signaling inhibitor, is a key regulator of bone remodeling [34]. Increased serum Dkk-1 was associated with a higher risk of bone destruction and osteoporosis in patients with RA [35]. The levels of Dkk-1 expression were elevated in the synovial fluid of RA patients [36,37]. Interestingly, the expression of Dkk-1 by synovial fibroblasts leads to the inhibition of osteoblast differentiation and new bone formation [38]. Understanding the crosstalk between the Wnt pathway and RANKL-associated bone effects may facilitate current pharmacologic developments. 

Wnt signaling is also regulated by proinflammatory pathways, namely TNF-α and IL-1 β (through the induction of Dkk-1 and sclerostin), IL-6, and IL-17 [39]. Additionally, Wnt signaling is inhibited by plasmablasts and plasma cells through the expression of Dkk-1 [34,39], which influences the expression of RANKL. Studies on this are still in the laboratory setting without further confirmation in patients with RA.

### 2.2. Bone Effect of Biological/Targeted Therapies in Rheumatoid Arthritis

Effective treatment is necessary for the bothersome symptoms related to RA bone involvement. According to European League Against Rheumatism (EULAR) recommendations, synthetic and biologic disease-modifying antirheumatic drugs (DMARDs) are the mainstay in RA treatment [40]. Apart from conventional synthetic DMARDs (methotrexate, leflunomide, sulfasalazine, and hydroxychloroquine) and glucocorticoids, the development of biological (b) DMARDs and targeted synthetic (ts) DMARDs has brought RA management into a new era. In clinical trials, the clinical performance of biologics and target therapies were favorable, with tolerable adverse effects.

B/tsDMARD therapy directly targets or indirectly modulates cytokines and halts the inflammatory cascade; moreover, the pathways that biologics targets also interact with the innate and adaptive immune system. The application of b/tsDMARD therapies has demonstrated anti-inflammatory effects for RA patients. Since the advent of b/tsDMARDs, the regimens have been found to alleviate clinical symptoms, improve the quality of life, and decelerate joint damage. To date, cumulative evidence has discussed the effect of b/tsDMARDs on bone metabolism, and we summarized its pathophysiology in Figure 1. 

#### 2.2.1. Tumor Necrosis Factor Inhibitors

Tumor necrosis factor (TNF) is a protein that is mainly produced by macrophages or monocytes, which participates in immune response regulation [14]. TNF-α plays an essential role in RA, which affects the action of synovial cells, macrophages, T-cells, B-cells, and endothelial cells [14]. The binding of TNF and TNF receptors activates caspase-dependent death signaling pathways with anti-apoptotic and proinflammatory responses [41]. TNF inhibitors are engineered monoclonal antibodies developed to competitively bind to TNF receptors and induce cell lysis by activation of complement-dependent or antibody-dependent cellular cytotoxicity [41]. The effects of TNF inhibitors (adalimumab, certolizumab pegol, etanercept, golimumab, and infliximab) on bone mineral density and osteoporosis in RA patients have been studied. TNF blockades may slow down generalized bone resorption, in association with clinical improvements [42]. In mouse models, inhibition of TNF showed a positive effect on bone formation and decreased osteoclastogenesis [43]. Anti-TNF therapy was also associated with elevated bone formation markers (e.g., procollagen Type I N-terminal propeptide (PINP)) and a decrease in bone resorption markers (e.g., C-terminal telopeptide of Type I collagen (CTX-I) and C-terminal cross-linked telopeptide of Type I collagen (ICTP)) in serum [44,45]. The effects on bone remodeling produced decreased levels of DKK-1 and increased levels of intact PINP [46]. A systematic review concluded that TNF inhibition might have a protective effect on the cartilage in the joint microenvironment [47].

In clinical observations, adalimumab has the potential to reduce bone damage and halt hand bone loss [48,49]. Infliximab has been found not only to increase BMD [50] but also to prevent arthritis-related osteoporosis and suppress tendon inflammation, thus alleviating tendon-related pain [51]. In patients with RA treated with infliximab, spine and hip bone damage was arrested, whereas metacarpal cortical hand bone destruction was not stopped [52]. Nevertheless, a recent study concluded that long-term use of infliximab did not affect bone microstructure and morphology in rats in the absence of an inflammatory condition [53]. Moreover, TNF inhibitors may suppress joint destruction and reduce the joint soreness caused by synovitis in a study with a 12-month follow-up period [47]. TNF inhibitors provoked a short-term rise in PTH levels and an early increase in bone turnover [54,55]. However, the evidence regarding fracture risk among RA patients receiving biologics is conflicting [56,57].

#### 2.2.2. Abatacept

Abatacept is a cytotoxic T lymphocyte-associated antigen 4 immunoglobulin fusion protein (CTLA-4-Ig) that can bind to CD80 or CD86, and subsequently prevent the signaling between T-cells and antigen-presenting cells [58]. Abatacept also competes with CD28 for CD80 or CD86 binding, and selectively regulates T-cell activation [59]. The inhibition of CD28-mediated T-cell activation effectively controls inflammation and inhibits bone erosion during RA [60]. By interfering with intracellular calcium oscillations in bone marrow macrophages, abatacept directly inhibits osteoclastogenesis [61]. Additionally, abatacept promoted T-cell Wnt protein production and prevented bone loss in a mouse model [62,63].

Clinically, serum OPG was significantly elevated and serum Dkk-1 was considerably lower 6 months after the introduction of abatacept [64]. The efficacy of abatacept for increasing BMD at the femoral neck was better than that of other bDMARDs [65]. Abatacept also had good efficacy for improving BMD at the femoral neck in patients with RA [66]. In RA patients with more severe disease activity and higher anti-CCP2 concentrations, treatment with abatacept was associated with more significant improvements in patient-reported outcomes over the following 6 months [67]. Hence, abatacept may improve RA patients’ quality of life and daily activity through mitigating bone pain and joint pain.

#### 2.2.3. Rituximab

Rituximab is a monoclonal antibody against CD20 that selectively targets B-cells and is approved as a second-line therapy for RA patients [68,69]. Since CD20 molecules are involved in complement activation, rituximab therefore induces complement-mediated cytotoxicity [68]. Rituximab may also cause structural changes and apoptosis of inflammatory cells [68,69]. Such B-cell depletion therapy has been reported to have a direct effect on bone remodeling in mouse models [70]. Rituximab and its associated immune response may have an essential role in regulating osteoblasts and osteoclasts [71,72]. Moreover, rituximab has the benefit of abrogating joint destruction in RA by inhibiting osteoclastogenesis [73]. Rituximab treatment is reported to be associated with the suppression of synovial osteoclast precursors and RANKL expression and a decrease in the serum RANKL/OPG ratio [73].

In a clinical study, rituximab treatment was associated with a significant improvement in femoral BMD. The application of rituximab further reduced bone pain and prevented patients from developing osteoporosis or fractures. Additionally, there was a substantial increase in P1NP and bone-specific alkaline phosphatase (BAP) [74]. Future rigorous trials are needed to provide solid evidence of the association between clinical symptoms and the use of rituximab.

#### 2.2.4. Tocilizumab

IL-6, a glycopeptide whose gene is located on chromosome 7, is involved in both T-cell and B-cell proliferation and differentiation [75]. Tocilizumab is a monoclonal antibody that binds to soluble and membrane-bound IL-6 receptors, promoting the inhibition of IL-6R signal transduction to inflammatory mediators [75,76]. IL-6 inhibitors also prevent bone damage and cartilage degeneration in RA patients [77,78]. Furthermore, IL-6 inhibition retards bone loss progression independently of its impact on disease activity [79]. In mouse models of collagen-induced and antigen-induced arthritis, IL-6 inhibition slowed down the progression of arthritis but did not ameliorate arthritis [80]. IL-6 inhibitor management also prevented mechanical hyperalgesia and suppressed calcitonin gene-related peptide (CGRP) expression in osteoporotic models [81].

In a 1-year prospective study, RA patients receiving tocilizumab exhibited a decrease in serum DKK-1 concentrations and an increase in bone formation markers without a significant change in BMD [82]. In contrast, tocilizumab increased total hip BMD with denosumab therapy for osteoporotic patients with RA [83]. Tocilizumab also stabilized BMD in a multicenter single-arm study [84] and increased the BMD of patients who had osteopenia at baseline [85]. In ACPA-positive patients, 2 years of tocilizumab treatment reduced bone resorption and increased femoral BMD [86].

#### 2.2.5. Janus Kinase Inhibitors

Janus kinase (JAK) inhibitors are small molecules that inhibit the JAK family enzymes (i.e., JAK1, JAK2, JAK3, and tyrosine kinase 2) that have a crucial role in the cell signaling processes leading to the immune and inflammation responses observed in RA [87,88]. JAK inhibitors induced bone repair by altering gene expression and increasing the activity of osteoblasts, supporting the use of inhibitors as potential anabolics [89]. Baricitinib, a selective inhibitor of JAK1 and JAK2, has been reported to have osteoprotective effects, increasing mineralization in bone-forming cells [90,91]. Osteoclastogenesis was also said to be suppressed by baricitinib via reducing RANKL expression in osteoblasts [91]. Baricitinib (4 mg once daily) in patients with moderate to severe RA inhibited the progression of radiographic joint destruction and relieved clinical symptoms such as joint swelling and tenderness. In addition, the application of tofacitinib in RA has been shown to affect osteoclasts directly and to inhibit osteoclast differentiation and proliferation [92]. Tofacitinib was also clinically effective in stabilizing BMD and lowering multiple bone markers such as P1NP and RANKL [93].

## 3. Spondyloarthritis 

### 3.1. Bone Remodeling and Osteoporosis in Spondyloarthritis

Pathological bone formation is one of the most iconic hallmarks of spondyloarthritis (SpA), including axial-SpA and psoriatic arthritis (PsA) [94]. In patients suffering from long-term axial-SpA, bone formations that manifest as entheseal bone formation, periostitis, and spinal syndesmophytes are strongly correlated with the burden of the disease and resulting disability [94]. The adverse effects caused by bone losses in spondyloarthritis may impair the patients’ quality of life and activities of daily living [94]. Previous observational studies have evaluated the association between SpA and osteopenia or osteoporosis, with conflicting results [95]. A previous meta-analysis concluded that patients with psoriatic diseases have a greater risk of developing fragility fractures compared with controls [96]. However, this higher risk of fractures may not necessarily correlate with lower BMD or a higher risk of osteoporosis [96]. Another meta-analysis provided more in-depth evidence about the bone microstructure and bone strength in psoriatic disease [97]. Psoriatic patients had lower volumetric BMD than non-psoriatic controls. On the other hand, osteoporosis is a frequent complication in patients with ankylosing spondylitis (AS) [94]. Low BMD, osteoporosis, and fractures are reported to be associated with AS [98]. Bone health has proven to be a crucial issue in patients with SpA [94,95]. The following paragraphs further describe the roles of distinct pathways and biologic therapeutics in bone metabolism. 

#### 3.1.1. Proinflammatory Cytokines

Proinflammatory cytokines (such as TNF-α and IL-17) are involved in the bone remodeling process of SpA [99]. Extensive experimental and clinical evidence has linked TNF-α to osteoclast development, but the role of osteoblast formation has remained somewhat controversial [100]. Surprisingly, the unique bone phenotype that occurs in PsA and AS coexists with both systemic bone destruction and new bone formation, which is likely to be the result of the actions of IL-23 and/or IL-17 on osteocytes [101,102]. IL-17, the prevailing inflammatory cytokine in many SpA patients, has been found to promote not only osteoclasts for bone resorption but also osteoblasts for bone formation [103,104]. IL-23 is another distinctive cytokine that is abundantly present in the affected skin or joints of PsA patients [103]. Generally, IL-23 is overexpressed in PsA, resulting in IL-22 upregulation and osteoblast-related gene induction. This process eventually contributes to both osteoblast expansion and enthesophyte formation [105,106].

#### 3.1.2. Autoantibodies

As RANKL and OPG play a pivotal role in the formation of osteoporosis, studies on autoantibodies against OPG have provided further knowledge. In a recent cohort, anti-OPG antibodies have been isolated in SpA patients, and a correlation with low BMD values and fractures was found [107,108]. In patients with RA, OPG levels were elevated and independently associated with disease severity; that is, OPG levels were higher in severe RA than in mild RA [109]. Moreover, genetic analysis has suggested that the OPG SNP haplotype was associated with HLA-B27 negativity in AS patients [110].

#### 3.1.3. The Wnt Signaling Pathway and Dkk-1

Dkk-1 is involved in the Wnt signaling pathway and also has evidence regarding bone metabolism in patients with SpA. In a study conducted by Rossini et al., Dkk-1 levels were found to correlate with low BMD and the prevalence of vertebral fractures among AS patients [111]. Nevertheless, Dkk-1 serum levels were inversely correlated with lumbar spine Z-score BMD, and higher serum levels of Dkk-1 were associated with a higher prevalence of one or more vertebral fractures. An association between Dkk-1 and PTH was observed, and higher levels of PTH and lower levels of Dkk-1 were also measured in AS patients. Future clinical and animal studies are warranted to investigate the Wnt signaling pathway and bone involvement in SpA. 

### 3.2. Effects on Bone of Biological/Targeted Therapies in Spondyloarthritis

Apart from low back pain, patients with SpA may experience different degrees of bone pain or bone loss in their entire disease course. Evidence-based recommendations for the treatment of axial SpA and PsA were updated in 2019 [112,113]. Targeted therapies against TNF, IL-17, IL-23, and the downstream pathways appear to be of significant clinical meaning according to the promising therapeutic results. However, the exact underlying mechanism regarding the application of these biologics in the inflammatory process is not yet clear [114]. Whether these novel biologics alleviate bone symptoms in different clinical presentations of SpA patients has also not been elucidated. The effects of biologics in SpA on bone mineral density have been reported, but the existing literature has been far less than that of RA. The impact of biologic therapies on bone metabolism is illustrated in Figure 1.

#### 3.2.1. Tumor Necrosis Factor Inhibitors

TNF inhibitors, such as adalimumab, certolizumab pegol, etanercept, golimumab, and infliximab, have been utilized in the management of SpA. Preclinical research had indicated that TNFα inhibitors could increase collagen synthesis in osteoblasts and inhibit osteoclast production [115]. In synovial fibroblasts, TNF inhibitors may also downregulate angiogenesis by activating the transcription factors and the NF-κB signal transduction pathway [116]. In patients with AS, the application of TNF-targeting therapies has been reported to slow radiographic progression and reduce disease activity in recent observational studies [117]. Lumbar and hip BMD have improved after using TNF inhibitors [118]. Furthermore, TNF inhibitors also decrease CRP and act inversely on Dkk-1 and SOST in patients with AS [47]. A recent meta-analysis concluded that TNF inhibitor use for more than 4 years was associated with delayed structural progression [119]. Joint tenderness may be alleviated by TNF inhibitors due to their effects on reducing synovial inflammation [116].

PsA-related bone diseases were better controlled by bDMARDs such as TNF inhibitors [120]. A post-hoc analysis of adalimumab in PsA revealed that the inhibition of radiographic destruction was greater than expected in the control rather than in clinical disease activity alone [121]. Another open-label study of adalimumab also showed similar results, namely that the improvement in PsA disease activity was not correlated with that of bone erosion [122]. Moreover, adalimumab improves enthesitis and may reduce joint aches as well as enhancing the range of motion in patients with PsA in magnetic resonance imaging [123]. By contrast, bone density measured by peripheral quantitative computed tomography did not show significant changes after treatment with TNF inhibitors [124].

#### 3.2.2. IL-17 Inhibitors

IL-17A, a member of the IL-17 superfamily of cytokines, is known to be involved in the pathomechanisms of SpA manifestations in skin, joints, and entheses. As evident in synovial samples, IL-17A signaling pathways are related to the actions of natural killer cells, tissue-resident memory T-cells, and innate lymphoid cells [125]. Secukinumab, ixekizumab, and brodalumab are IL-17-targeting therapeutical options. Their clinical performance is favorable, and the adverse effects seem to be tolerable. Both short-term and long-term clinical studies have indicated that patients treated with IL-17A inhibitors might reduce synovial inflammation and the destruction in bony structures versus a placebo [126]. Compelling evidence to date has shown that secukinumab improves arthralgia, swelling, physical function, and quality of life in SpA patients [126]. The PSARTROS study demonstrated that secukinumab used in psoriatic patients for 24 weeks showed no progression of catabolic or anabolic joint alterations [127,128]. The functional strength remained stable with a reduction in disease activity as measured by Disease Activity in PSoriatic Arthritis [128]. Experimentally, IL-17A promoted local mesenchymal stem cells to differentiate into osteoblasts and also increased mineralization in AS [129]. In an animal model of pathogenic SpA, treatment with anti-IL-17A delayed the effect of bone loss and reduced pathological bone formation [130]. 

#### 3.2.3. Janus Kinase Inhibitors

JAK inhibitors are novel biologics for patients with SpA and have been studied in several clinical trials. Nevertheless, data regarding the effect of JAK inhibitors on bone resorption are scarce in SpA. The rationale for using JAK inhibitors in SpA came from the inhibition of various signaling pathways [131]. As JAK-dependent cytokines are involved in the pathogenesis of SpA, including IFNγ, IL-7, IL-12, IL-15, IL-22, and IL-23, blockade of the JAK-STAT pathway may inhibit the cellular function of a broad range of innate and adaptive cell types in SpA [131,132]. JAK inhibitors have been found to significantly reduce the secretion of the proinflammatory mediators MCP-1 and IL-6 in ex vivo studies [132]. The effect of JAK inhibitors on reducing inflammation in arthritis is promising; however, the data on bone mineral density are limited. More evidence is necessary to elucidate the clinical implications of JAK inhibitors for patients with SpA.

#### 3.2.4. IL-23 Inhibitors

IL-23, a proinflammatory cytokine associated with the production of IL-17, IL-22, and TNF, was found to play a vital role in both innate and adaptive immunity [133]. The potential pathophysiologic effects of IL-23 in chronic inflammatory diseases have been identified in animal studies [134]. Data from preclinical IL-23 knockout models demonstrated the importance of IL-23 in the pathogenesis of arthritis [135]. Moreover, evidence from genetic analyses has suggested an association between the IL-23/IL-17 pathway and AS [136]. However, these findings in laboratory settings did not generally translate into therapeutic effects in SpA patients [137]. Two IL-23 inhibitors, risankizumab (the p19IL-23 inhibitor) and ustekinumab (the p40IL-12/23 inhibitor), did not achieve the primary endpoint in AS trials [138,139]. 

The evidence on whether the net effect of IL-23 on bone turnover is conflicting [140]. With regard to the catabolic effects, IL-23 has been suggested to promote osteoclastogenesis in human cells independently of the osteoblasts or exogenous soluble RANKL. In experimental settings, IL-23 was found to induce the activation of precursor cells and the associated proteins in RANKL-mediated osteoclastic differentiation [141,142]. On the other hand, genetic data have also reported potential anabolic effects in IL-23-related pathways. Genetic analysis from mice has suggested protective effects on bone. IL-23 can also induce the production of granulocyte-macrophage colony-stimulating factor (GM-CSF), an inhibitor of osteoclast differentiation [143]. However, this effect is mediated by TH17 cells and the subsequent production of IL-17, suggesting a net catabolic effect on bone metabolism towards IL-23. Trials of ustekinumab in patients with PsA, which demonstrated inhibition of the progression of bone destruction [144], provided clinical evidence of this debate. Research on the associations of IL-23 with other proinflammatory cytokines and mesenchymal cells is ongoing to understand their effects on bone metabolism better.

#### 3.2.5. PDE4 Inhibitors

Phosphodiesterase 4 (PDE4) is an enzyme in the process of cyclic adenosine monophosphate (cAMP) degradation, which involves numerous biologic responses in human cells [145]. Previous literature has suggested the pathogenic role of cAMP inhibition on diseases such as chronic obstructive pulmonary disorder, inflammatory bowel disease, and PsA [146]. As PDE4 is involved in these inflammatory processes, inhibition of PDE4 may provide profound anti-inflammatory properties. 

Apremilast, an oral PDE4 inhibitor, was shown to have a good safety profile, and its clinical implications have been investigated in recent trials. Apremilast blocks several pathways involving proinflammatory cytokines and chemokines, such as TNF-α, IL-23, CXCL9, and CXCL10 [145]. In contrast to the biologics mentioned above, which neutralize proinflammatory factors at the protein level, apremilast modulates these factors at the mRNA level [147]. 

PDE4 inhibitors may be associated with the inhibition of osteoclastogenesis in inflammatory arthritis. An ex vivo study using synovial fluid mononuclear cells of PsA patients suggested the inhibitory role of apremilast on IL-12/IL-23p40 [145]. Apremilast could also inhibit osteoclast fusion molecules such as dendritic cell-specific transmembrane protein and osteoclast-specific transmembrane protein, preventing osteoclastogenesis [146]. Studies on another PDE4 inhibitor, rolipram, have shown a potential blocking effect on PTH-induced osteoclast formation via the inhibition of calcitonin [147]. Additionally, the effect of apremilast on IL17A-mediated osteoclastogenesis has also been indicated by experimental studies using human peripheral blood mononuclear cells [147]. More in-depth and clinical evidence may improve our knowledge of the protective effects of PDE4 inhibitors on bone destruction in inflammatory arthritis. 

## 4. Conclusions

In this narrative review, we have discussed the recent understanding of the pathogenic effects on bone in RA and SpA. The crosstalk between inflammation and bone metabolism is complicated. Proinflammatory cytokines, autoantibodies, and multiple signaling pathways play essential roles in bone destruction in RA and SpA patients. Current evidence has suggested the effect of novel therapeutics for improving clinical symptoms as well as enhancing the quality of life and daily activities of patients with RA and SpA. Biological and targeted therapies have provided alternative options for managing RA and SpA. However, the effects of these therapies on bone are not fully understood. As the positive effects on bone mineral density and bone metabolism of inflammatory arthritis are reviewed here, more research is warranted on the underlying mechanisms and their clinical impact. Future research is encouraged to demonstrate how these biological and targeted therapies best mitigate bone destruction in chronic inflammatory arthritis.

## Figures and Tables

**Figure 1 ijms-23-04111-f001:**
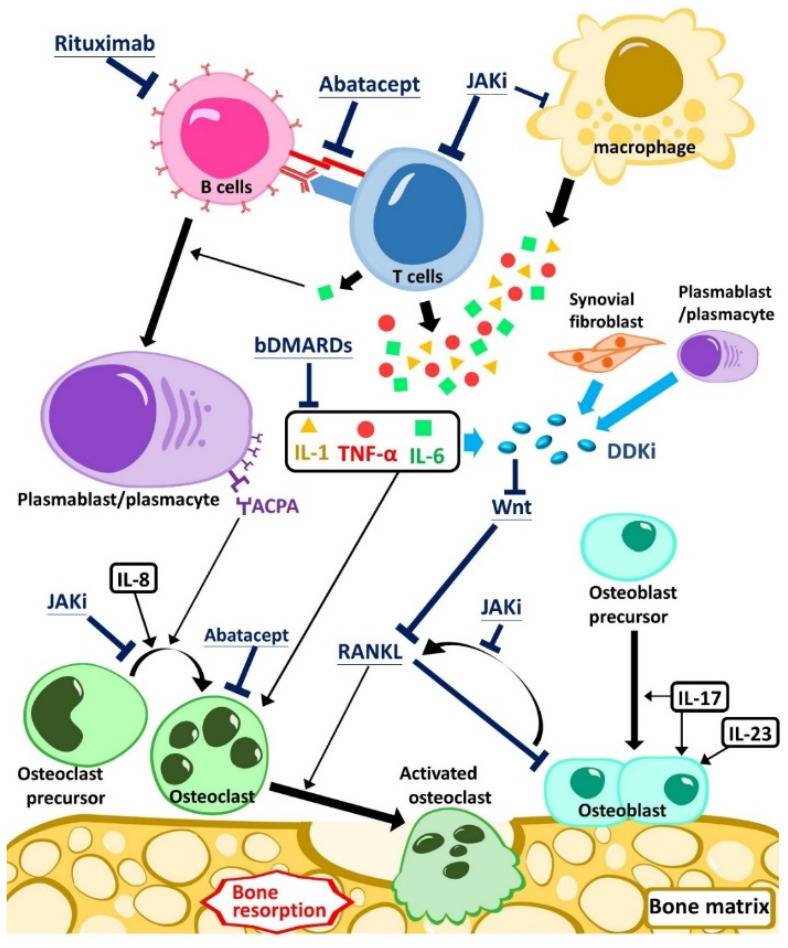
The mechanism of the effect of biological/targeted therapies on bone metabolism in inflammatory arthritis. bDAMARDs, biologic disease-modifying antirheumatic drugs (DMARDs); IL, interleukin; JAKi, Janus kinase inhibitors; TNF-α, tumor necrosis factor-α.

## Data Availability

This review presented data that were online for the public.

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
