# Peer review of "Effects of Biological/Targeted Therapies on Bone Mineral Density in Inflammatory Arthritis"

_ijms, 2022, doi:10.3390/ijms23084111_

Round 1

Reviewer 1 Report

In the current review, the authors presented the pathophysiology of rheumatoid arthritis and spondyloarthritis. They focused also on the effect of targeted therapies, including biological and small molecule inhibitors, on bone mineral density and osteoporosis of inflammatory arthritis.

Comments and suggestions:

-pg 1, line 39-40: you can’t write that “you reviewed the pathophysiology of several inflammatory arthritis” since you treated only rheumatoid arthritis and spondyloarthritis. It’s not appropriate “several”. Please reformulate.

-In the abstract you wrote: “Recent progress in both animal and clinical studies has provided a better understanding.” In the article you just mention such studies. They need to be detailed.  The article is too short and presents too little information for a review article - pg 3, point 2.2.1: please explain more clearly the mechanism of action for the  TNF inhibitors -It’s necessary to present a more detailed mechanism of action for each class of drugs (where are currently known). Please add

- Points 2.2 and 3.2: For each class of therapies given, please highlight what symptoms have been improved in order to emphasize the effectiveness of the therapy.

- The place of Figure 1 is not at the conclusions. Please move it to another part of the article with some additional explanations. Also, at conclusion please underline the importance of these therapies in improving the quality of life  and activities of daily living for patients.  

Author Response

Responses to Comments from Reviewer 1

Comment 1:

pg 1, line 39-40: you can’t write that “you reviewed the pathophysiology of several inflammatory arthritis” since you treated only rheumatoid arthritis and spondyloarthritis. It’s not appropriate “several”. Please reformulate.

Response 1:

We thank the reviewer for this suggestion. We removed the word “several” and specified more clearly about the inflammatory arthritis we reviewed in this article (Page 2, Lines 52-53).

Comment 2:

In the abstract you wrote: “Recent progress in both animal and clinical studies has provided a better understanding.” In the article you just mention such studies. They need to be detailed.  The article is too short and presents too little information for a review article - pg 3, point 2.2.1: please explain more clearly the mechanism of action for the TNF inhibitors -It’s necessary to present a more detailed mechanism of action for each class of drugs (where are currently known). Please add.

Response 2:

In response to the reviewer’s comment, we have incorporated more explanation to the mechanism of action for TNF inhibitors and present detailed information for each class of drugs (Point 2.2 and Point 3.2). Moreover, the abstract and introduction were also extended to provide detailed information about our review article. We appreciate the reviewer’s time and effort in enhancing the quality of our work.

Comment 3:

Points 2.2 and 3.2: For each class of therapies given, please highlight what symptoms have been improved in order to emphasize the effectiveness of the therapy.

Response 3:

This suggestion is excellent and has been well taken. We have added sentences to highlight the symptoms improved after the application of each class of therapies (Point 2.2 and Point 3.2).

Comment 4:

The place of Figure 1 is not at the conclusions. Please move it to another part of the article with some additional explanations. Also, at conclusion please underline the importance of these therapies in improving the quality of life and activities of daily living for patients.

Response 4:

In response to the reviewer’s suggestion, we moved Figure 1 to Introduction to provide a first impression of our discussion on the mechanisms. We have also added the importance of improving the quality of life and activities of daily living for patients under the therapies in the Conclusion.

Reviewer 2 Report

Referee’s comment

Article n° IJMS 164175

Title: Effects of Biological/Targeted Therapies on Bone Mineral Den- sity in Inflammatory Arthritis

Authors:  Chen T-L,  Chang  K-H,  Su K-Y.

General comment

The present review is of interest and well-organized. The aims and objectives are clear, the hypothesis is sound and the data are well presented. The abstract well reflects the content of the article. The introduction describes what the author hoped to achieve accurately, and clearly states the problem being investigated. The paper should be of interest to scientists working in the field of rheumatoid arthritis as well as others with closely related research interests. Finally, the article is sufficiently novel to warrant the present publication. However,  I have only a  suggestion to make the paper more suitable and interesting.

1) Arthritic pathology should also be considered in the context of several factors that converge together in the progression of damage. The reduced mineral density of the bone also seems to be due to a reduced thickness of the cartilage. Therefore, the contribution of cartilage to bone metabolism should also be considered. Consequently, also drugs that directly promote cartilage growth and indirectly improve the bone component. I think it is important to consider this aspect and insert it into the text.

Author Response

Responses to Comments from Reviewer 2

Comment 1:

The present review is of interest and well-organized. The aims and objectives are clear, the hypothesis is sound and the data are well presented. The abstract well reflects the content of the article. The introduction describes what the author hoped to achieve accurately, and clearly states the problem being investigated. The paper should be of interest to scientists working in the field of rheumatoid arthritis as well as others with closely related research interests. Finally, the article is sufficiently novel to warrant the present publication.

Response 1:

We thank the reviewer for the positive feedback. We also appreciate the reviewer’s time and effort in enhancing the quality of our work.

Comment 2:

I have only a suggestion to make the paper more suitable and interesting. Arthritic pathology should also be considered in the context of several factors that converge together in the progression of damage. The reduced mineral density of the bone also seems to be due to a reduced thickness of the cartilage. Therefore, the contribution of cartilage to bone metabolism should also be considered. Consequently, also drugs that directly promote cartilage growth and indirectly improve the bone component. I think it is important to consider this aspect and insert it into the text.

Response 2:

This suggestion is excellent and has been well taken. The contribution of cartilage to bone metabolism has been inserted into the text.

Reviewer 3 Report

The paper is a narrative review of the effect of biological and targeted synthetic DMARDs when used for the treatment of inflammatory arthritides on bone mineral density.

Although limited by its narrative design the paper is interesting, however I have the following comments for the authors:

  • Please consider to improve the abstract.
  • Page 1, lines 35-37: the sentence is not clear.
  • Page 1, line 38 “has” not "have"?
  • Page 1, line 39 “review […] reviewed”, please try to avoid repetition
  • Page 2, lines 50-51. Please provide appropriate references for the two sentences.
  • Page 2, line 58. Please define TNF.
  • Page 2, lines 74-75. Please provide appropriate references.
  • Page 2, lines 86-88. Please provide appropriate references.
  • Page 2, lines 90-93. Please provide appropriate references for the two sentences.
  • Page 3, lines 109-111. Please provide appropriate references.
  • Page 4, lines 163-165. How does this sentence relate to the topic of the paper?
  • Page 4, line 166. Please correct “Rrituximab”
  • Page 5, lines 204-207. How does this sentence relate to the topic of the paper?
  • Page 5, lines 213-219. Please provide appropriate references for the sentences.
  • Page 5, lines 224-227. Please provide appropriate references for the sentences.
  • Page 6, lines 246-247. Please rewrite the sentence so that it is not redundant with the previous one.
  • Page 6, lines 248-250. How does this sentence relate to the topic of the paper?
  • Page 6, section 3.2.1. This section report very little information regarding TNFi and bone density in SpA, please consider to include appropriate studies to the discussion (e.g.: 10.1186/s12891-021-04708-5; 10.1002/acr.22519; 10.1186/s13075-019-1938-3).
  • Page 6, line 273. Please define "TNFi"
  • Page 7, line 297. Please eliminate the line.
  • Please correct the references.

Author Response

Responses to Comments from Reviewer 3

We thank the reviewer for the extensive assessment of our manuscript, and for important and helpful comments. We also appreciate the reviewer’s time and effort in enhancing the quality of our work.

Comment 1: Please consider to improve the abstract.

Response 1: We thank the reviewer for this excellent suggestion. We have added sentences to incorporate more comprehensive overview of our article in the Abstract (Page 1, Lines 16-29).

Comment 2: Page 1, lines 35-37: the sentence is not clear.

Response 2: In response to the reviewer’s suggestion, we have made the sentences more clear to emphasize the main purpose of this narrative review (Page 2, Lines 48-52).

Comment 3: Page 1, line 38 “has” not "have"?

Response 3: This sentence has been modified.

Comment 4: Page 1, line 39 “review […] reviewed”, please try to avoid repetition

Response 4: We thank the reviewer for this careful check. The sentence has been corrected (Page 2, Lines 56).

Comment 5: Page 2, lines 50-51. Please provide appropriate references for the two sentences.

Response 5: We have provided the references for these two sentences (Page 3, Lines 71-72).

Comment 6: Page 2, line 58. Please define TNF.

Response 6: TNF has been defined (Page 3, Line 79).

Comment 7: Page 2, lines 74-75. Please provide appropriate references.

Response 7: We have provided the references for the sentence (Page 3, Lines 97).

Comment 8: Page 2, lines 86-88. Please provide appropriate references.

Response 8: We have provided the references for these sentences (Page 3, Lines 109, 112).

Comment 9: Page 2, lines 90-93. Please provide appropriate references for the two sentences.

Response 9: We have provided the references for these two sentences (Page 3, Line 113; Page 4, Line115).

Comment 10: Page 3, lines 109-111. Please provide appropriate references.

Response 10: We have provided the references for the sentence (Page 4, Line 133).

Comment 11: Page 4, lines 163-165. How does this sentence relate to the topic of the paper?

Response 11: The Wnt signaling may influence RANKL expression. We added more explanation for clarity (Page 4, Line 133).

Comment 12: Page 4, line 166. Please correct “Rrituximab”

Response 12: We thank the reviewer for the spelling check. We corrected this word (Page 5, Line 204).

Comment 13: Page 5, lines 204-207. How does this sentence relate to the topic of the paper?

Response 13: We have modified the sentences to avoid confusion (Page 6, Lines 247-253).

Comment 14: Page 5, lines 213-219. Please provide appropriate references for the sentences.

Response 14: We have provided the references for the sentences (Page 6, Lines 267-264).

Comment 15: Page 5, lines 224-227. Please provide appropriate references for the sentences.

Response 15: We have provided the references for the sentences (Page 7, Lines 270-272).

Comment 16: Page 6, lines 246-247. Please rewrite the sentence so that it is not redundant with the previous one.

Response 16: This sentence is redundant and we considered to remove it (Page 7, Line 291).

Comment 17: Page 6, lines 248-250. How does this sentence relate to the topic of the paper?

Response 17: We have modified the sentence for clarity (Page 7, Lines 293-294).

Comment 18: Page 6, section 3.2.1. This section report very little information regarding TNFi and bone density in SpA, please consider to include appropriate studies to the discussion (e.g.: 10.1186/s12891-021-04708-5; 10.1002/acr.22519; 10.1186/s13075-019-1938-3).

Response 18: In response to the reviewer’s suggestion, we have added sentences and modified the references (Point 3.2.1, Reference 118, 120, 124).

Comment 19: Page 6, line 273. Please define "TNFi"

Response 19: In response to the reviewer’s suggestion, we have defined TNFi and modified the sentences for clarity (Point 3.2.1).

Comment 20: Page 7, line 297. Please eliminate the line.

Response 20: We eliminated this line for redundancy.

Comment 21: Please correct the references.

Response 21: In response to the reviewer’s suggestion, we have corrected the references to avoid duplicate.

Reviewer 4 Report

Dear Authors,

Please improve and extent your introduction

Please add more conclusion to your review

You have mistakes in references. Appear two groups of references

Please tell us if is it a narrative or systematic review. You don’t say us which is the inclusion and exclusion criteria for selection of the articles. How many articles did you select and include in the review.

Kind regards

Author Response

Responses to Comments from Reviewer 4

Comment 1:

Please improve and extent your introduction.

Response 1:

We thank the reviewer for this excellent suggestion. We have added sentences to incorporate more introduction to our article (Page 2, Lines 33-60).

Comment 2:

Please add more conclusion to your review. 

Response 2:

In response to the reviewer’s suggestion, we have improved and extended the conclusion (Page 10, Lines 410-421).

Comment 3:

You have mistakes in references. Appear two groups of references

Response 3:

We thank the review for the careful check. The references have been checked and revised to accurate form.

Comment 4:

Please tell us if is it a narrative or systematic review. You don’t say us which is the inclusion and exclusion criteria for selection of the articles. How many articles did you select and include in the review (Page 1, Line 20; Page 2, Line 52).

Response 4:

In response to the reviewer’s suggestion, the article type (narrative review) has been specified in the Abstract and Introduction.

Round 2

Reviewer 1 Report

In my opinion the article has been significantly improved but not all the requested changes were made throughout the text. So,

Concerning the affirmation: “Recent progress in both animal and clinical studies has provided a better understanding”, statement originally made in the abstract the authors decided to delete the statement instead to detail it. I’m not satisfied with this option.

It’s  not appropriate to write about Figure 1 at the conclusions.

After a minor revision the manuscript is suitable for publication.

Best regards,

Author Response

Responses to Comments from Reviewer 1

Comment 1:

Concerning the affirmation: “Recent progress in both animal and clinical studies has provided a better understanding”, statement originally made in the abstract the authors decided to delete the statement instead to detail it. I’m not satisfied with this option.

Response 1:

In response to the reviewer’s suggestion, we have added sentences to incorporate more in-depth introduction to the article in Abstract (Page 1, Lines 18-24).

Comment 2:

It’s not appropriate to write about Figure 1 at the conclusions.

Response 2:

In response to the reviewer’s comment, we have modified the first sentence in the Conclusion (Page 10, Lines 417-419).

Reviewer 3 Report

I have no further comment.

Author Response

Responses to Comments from Reviewer 3

Comment 1: I have no further comment.

Response 1: We thank the reviewer for the positive feedback. We also appreciate the reviewer’s time and effort in enhancing the quality of our work.